# Reflective Empiricism and Empirical Animal Ethics

**DOI:** 10.3390/ani12162143

**Published:** 2022-08-21

**Authors:** Hannah Winther

**Affiliations:** Department of Philosophy and Religious Studies, Norwegian University of Science and Technology, 7491 Trondheim, Norway; hannah.winther@ntnu.no

**Keywords:** empirical ethics, animal ethics, Cora Diamond, Iris Murdoch

## Abstract

**Simple Summary:**

A central methodological question in empirical ethics is how empirical research can inform normative analysis. This article attempts to answer this challenge by drawing on the philosophy of Iris Murdoch and Cora Diamond, defending Diamond’s concept reflective empiricism as an approach to empirical ethics. The article also discusses the relevance of these discussions for empirical animal ethics and argues that reflective empiricism can constitute a methodological approach in this field.

**Abstract:**

The past few decades have seen a turn to the empirical in applied ethics. This article makes two contributions to debates on this turn: one with regard to methodology and the other with regard to scope. First, it considers empirical bioethics, which arose out of a protest against abstract theorizing in moral philosophy and a call for more sensitivity to lived experience. Though by now an established field, methodological discussions are still centred around the question of how empirical research can inform normative analysis. This article proposes an answer to this question that is based on Iris Murdoch’s criticism of the fact/value distinction and Cora Diamond’s concept of reflective empiricism. Second, the discussion takes as a point of departure a study on genome-edited farmed salmon that uses qualitative research interviews and focus groups. Although there are several animal ethics studies based in empirical data, there are few works on the methodological challenges raised by empirical ethics in this area. The article contributes to these discussions by arguing that reflective empiricism can constitute a methodological approach to animal ethics.

## 1. The Empirical Turn in Bioethics

Most branches of philosophy have at some point undergone a turn (think, for example, of the linguistic turn, the experimental turn, the phenomenological turn or the ontological turn). The history of philosophy can be broken down into such turns, each the result of an attempt to revolt against the practices of previous philosophers and to remedy flaws and problems that held them back from progress.

Some turns have been classified as such retrospectively, to describe an important shift after it has occurred, while others are launched with fanfare to introduce a new school or method for doing philosophy. While it is difficult to say something general about what characterizes them, they share at least two traits. The first is an attempt to redefine or challenge existing conceptions about philosophy’s aim, scope and method. This involves delineating the field of what it is possible for philosophy to say something about and constructing frameworks that set standards for how we should progress our investigations and which tests have to be passed for something to count as knowledge. The second is that they are concerned with the relationship between knowledge and experience. Experience is such a bedrock concept in philosophy that its importance is easy to miss. However, questions such as what and how we can learn from experience and what standing experience should have when we make claims about the world continue to define what we think the appropriate methods for philosophy are.

Here I am concerned with a particular turn, namely the empirical turn in bioethics. It denotes a shift in bioethical methodology that has occurred in recent decades, where bioethicists have started to either conduct empirical research or borrow already existing data to inform normative reflection on bioethical issues. The concept of experience is at heart here too: its development has been prompted by the claim that bioethicists who merely rely on their own experience when they want to make normative claims about bioethical issues have an insufficient basis to make those claims and that empirical research, most commonly qualitative research in the form of interviews and focus groups, is a useful tool for acquiring a better basis.

As with any turn, discussions are concerned with its potential promise and practical feasibility. The main question is the following: How can qualitative research inform normative analysis in a meaningful way? “Inform” is a crucial word here, as the scope of the word can have different implications for what the empirical turn is supposed to involve. It can merely mean that we include facts in normative reasoning, but as Samia Hurst has pointed out, this is so trivial that it can hardly be taken to support the claim that a turn has occurred at all [1] (p. 440). Empirical bioethics has a taller order to fill, however, in its aspiration that empirically informed normative reflection should mean more than having facts to support argument. But what, exactly? The scope of this ambition varies and it can be challenging to get a grasp of exactly what it means. Empirical ethics aims to be both descriptive and normative [2] (p. 468), but how can this be achieved? “Paying attention to the empirical world means learning from it”, Ives et al. write and also “the empirical ought not to be a handmaiden to the normative (philosophical) monarch, it should be a partner” [3] (p. 6). This suggests the idea of an integration between the empirical and the normative, but what it consists in, and how it can be achieved, remains a contested topic.

In this article, I address this challenge by drawing on a set of theoretical resources outside of the ordinary scope of empirical bioethics literature, namely Cora Diamond and Iris Murdoch. I argue that these thinkers share empirical bioethics’ spirit and ambition to respect and illuminate moral experience and that they can offer useful resources for paying attention to experience in the way that contemporary empirical bioethics aims to (for a similar argument, see [4]).

The structure of the article is as follows. First, I briefly sketch the development that provoked bioethics’ turn to the empirical and discuss in further detail the integration problem briefly outlined above, using as a point of departure my own research on genome-edited farmed salmon. I proceed to offer an analysis of Murdoch’s discussion of facts and values that I believe provides a more adequate understanding of how we should understand that problem. To make this point, I discuss the article “Empirical data and moral theory: A plea for integrated empirical ethics” by Bert Molewijk, Anne M. Stiggelbout, Wilma Otten, Heleen M. Depuis and Job Kievit [5]. Even though there has been some time since this article was published, it is a good case study since it is a much cited and paradigmatic example of discussions of the integration problem in empirical bioethics. I argue that although the authors acknowledge the interdependence of facts and values, their conceptualization of the relationship between them is an expression of a metaphysical outlook that needs not be assumed in order to do empirical bioethics and which gives rise to the integration problem.

Second, I lay out an approach to empirical bioethics projects proceeding from Murdoch and Diamond, who both place themselves at odds with this metaphysical outlook. I discuss Diamond’s concept of *reflective empiricism*, which she describes as an empiricism of reflection on human experience, and argue that this can constitute a methodological approach to empirical bioethics. As an example, I consider Ives’ article “Encounters with experience” and his research on the concept of fatherhood, which I argue is exemplary of this kind of method. Finally, I discuss the relevance of this approach for animal ethics and argue that qualitative research can be useful to shed light on our moral relationships with animals.

## 2. Armchair Versus World

Writing in 1984, medical sociologist Renée Fox and Judith Swazey gave bioethics, a field that had gradually developed since the 1960s and was by then already an established discipline, a harsh verdict. Fox and Swazey had travelled to China to do fieldwork on the term ‘medical morality’ and obtain some cultural perspective on its American counterpart bioethics [6] (p. 336). On their account, the latter was severely flawed: its moral framework placed too much emphasis on logical reasoning and universal applicability, it assumed an approach where a general moral theory is applied and concepts derived from it based on codified methodological rules and techniques, and it was unable to accommodate values and relationships. Also, while the aim was to have bearings on ethically problematic issues, bioethicists rarely cared to examine the actual situations in which ethical problems occur. Instead they preferred, as Fox and Swazey put it, “ordered, cerebral, armchair inquiry” using thought experiments, and they preferred it because the formalistic data it generates are easier to fit into their moral frameworks [6] (pp. 335–356). As a result, bioethics distanced and abstracted itself from the complexity and ambiguity of actual cases, engaging in “intellectual and moral reductionism” [6] (p. 358).

Fox and Swazey were not alone in voicing this kind of criticism. Barry Hoffmaster, for example, complained that the analytically informed conception of philosophical morality in medical ethics “concentrates on developing and justifying theories and pays little attention to the practical utilization of those theories” [7] (p. 1156). Bioethics, briefly put, was accused of “practic[ing] a philosophical idealism in which the big problems are just that people think the wrong way” [8] (p. 650). These objections are commonly grouped under the heading *the social science critique*. The perception was that bioethics had become “too abstract, too general, too speculative, and too dogmatic” [9] (p. 2) by prioritizing idealized rational thought and excluding cultural and social factors.

Empirical bioethics emerged as a response to this critique. It seeks to develop a contextualized ethical analysis that is grounded in lived experience while also being critically normative. In one respect, this is not new—bioethics has always been concerned with the empirical in some sense. When bioethicists have wanted to discuss emerging biotechnologies, they have always cared to learn about how it works and they have also cared to collect relevant facts and use real cases. The empirical turn, instead, designates a methodological shift in which bioethicists started to borrow methods and data from the social sciences.

However, though many empirical bioethicists embraced the idea of conducting and using qualitative research, they remained unclear on exactly *how* empirical data should inform their philosophical analysis. Several methodologies and frameworks have been proposed, but currently, there is no consensus on what an appropriate methodology should look like [10] (p. 1). The question that such a methodology has to answer is this: how can we draw any conclusions about how we should act in a way that is meaningfully informed by people’s experiences?

## 3. Example I: Using Gene Editing Technologies on Farmed Salmon

These questions were foundational for the interdisciplinary research project I am connected with. This project takes as its departure the salmon farming industry in Norway, an important cornerstone in the new, blue economy. There are political goals for the growth of this industry, but at the same time, it faces challenges related to environmental impact and animal welfare. A few such examples include salmon lice and diseases, which severely impact the life quality of the farmed salmon, and the fact that they escape from the pens in large numbers, breeding with wild populations. However, there are high hopes that gene editing technologies such as CRISPR might contribute to solving these problems. Researchers have developed a sterile salmon [11], which, if it is put into production, can potentially solve the latter problem—one of the major environmental challenges for the industry.

These emerging possibilities raise pressing questions about the moral acceptability of using such technologies on farmed animals. How do we approach such questions as empirical bioethicists? In this project, we conducted in-depth research interviews with stakeholders, ranging from people who work directly with the fish on the fish farms, scientists who develop CRISPR technology and conduct experiments on fish, NGO representatives who oppose the industry from the outside and representatives from interest organizations for the protection of wild salmon and Samí culture and tradition (the Samí are the indigenous people who have their traditional settlements in the Arctic area, called Sápmi, which today encompasses parts of northern Norway, Sweden, Finland and the Kola Peninsula of Russia).

A reasonable question that might be raised is why we should bother to talk to these groups at all. The philosophical interest in this project is to say something about the moral acceptability of using CRISPR on farmed salmon. How can talking to stakeholders and citizens help us answer this question? After all, people can only tell us what their opinion about this normative question is. While some would argue that using gene editing technologies on animals amounts to playing God and infringes on the animals’ intrinsic value, others would point to the potential benefits and argue that gene editing is the natural next step in breeding, something which we have been doing for centuries anyway. The bioethicist should reach some normative conclusion and the role of these data in that discussion remains unclear.

I will leave this case for now and return to it later. Below, I consider what I take to be a common example of how the question of the relationship between the empirical and the normative is treated in empirical bioethics literature. This is the article “Empirical data and moral theory: A plea for integrated empirical ethics” by Molewijk et al. [5]. Up against their interpretation of facts and values as interdependent but separate domains, I defend Murdoch’s claim that values are in fact ubiquitous, and argue that this understanding has implications for empirical bioethical frameworks. While I concentrate on Molewijk and colleagues here, I believe that the current discussion will be of relevance for the broader discussion on the integration problem and attempts to address it. I will get back to these claims below, but first, I will show what Molewijk and his colleagues actually do in the paper.

## 4. Empirical Data and Moral Theory: Conceptualizations

In “Empirical data and moral theory: A plea for integrated empirical ethics”, the authors both propose a typology classifying different approaches to empirical bioethics and defend their own approach, termed *integrated empirical ethics*. Both undertakings are worth commenting on, and I will do so in order.

According to the typology, there are four kinds of empirical bioethicist researcher. The significant difference between them is the stance each of them take to the question of who the final arbiter should be in case of conflict; the moral theory or the morality within a social practice [5] (p. 56). *The prescriptive applied ethicist* holds that moral theory should change to better accommodate social practices. *The theorist* holds that the practices should be changed in light of the theory. *The critical applied ethicist* argues that there needs to be reciprocal adjustment, whereas *the particularist* thinks that there is not and should not be any interaction at all.

What these approaches share is the conceptualization of moral theory and empirical data as two different spheres that can be brought in and out of touch with each other. Consider, for example, the following few descriptions: Of a consequentialist approach to prescriptive applied ethics, Molewijk et al. write that they “need empirical data in order to determine which behavior is morally preferable” (Molewijk et al., 2004, p. 56). The theorist’s method is described as being “neither completely inductive nor completely deductive”; there is a “one way-interaction between moral theory and empirical data” [5] (p. 56), while the critical applied ethicists perceive a “mutual interaction between empirical data and moral theory” [5] (p. 57).

These descriptions reveal two assumptions. The first is that they seem to assume that the world can be divided into two, with morally neutral and describable facts on the one hand and our values and moral frameworks on the other. The second has to do with the role these data are supposed to play in our philosophical reflection: once the morally salient facts are identified, these data will be sorted into arguments about right and wrong. This paints a picture where the task of the empirical bioethicist is to collect data and use them to generate normative conclusions. Though this is a conception that few empirical bioethicists will want to support, it turns out to be challenging to shake it off completely. It is a conception that leads to the fact/value problem, which we take as a fundamental point of departure, whereas it in fact arises out of a particular metaphysical outlook.

That metaphysical outlook belongs to a kind of philosophy that both Murdoch and Diamond have consistently placed themselves at odds with. When I turn to these thinkers here, it is because I believe that their account gives a more adequate description of morality and our moral lives and that they offer methodological resources for empirical bioethics without falling prey to concerns about gaps between facts and values, ises and oughts. Murdoch’s criticism of the fact/value distinction, in particular, serves as a good gateway for discussing what they both find problematic with this outlook. I will therefore make room for a brief Murdoch interlude before we come back to Molewijk’s paper.

## 5. Interlude: Murdoch on Facts and Values

Though the fact/value dichotomy no longer has the standing it once did, and though the vast majority of philosophers either do not accept it as genuine or at least acknowledge its complexity, it continues to have an impactful legacy. It is therefore a dichotomy worth revisiting. Murdoch objected to the fact/value distinction at several points throughout her work. She was among the first to do so, as Diamond notes [12] (p. 79), though she is rarely credited for it. Murdoch might seem a somewhat dated example, since she developed her criticism in response to logical positivism, the reigning philosophical paradigm at her time. However, she still has something to teach us, and though the ideas she opposes have dwindled, they continue to shape discourse.

When Murdoch wrote, A. J. Ayer’s *Truth, Language, and Logic* was making philosophical waves. Taking his cue from Hume, Ayer maintained a sharp distinction between facts, which we can make meaningful statements about, and values, which is something that can be ascribed to facts. It is the former category that philosophers should be concerned with, Ayer argued, reducing ethics to the expression of subjective emotions and banishing metaphysics to the dark corners of meaninglessness. This view set the philosophical scene for Oxford in the 1950s and Murdoch took issue both with the distinction itself and the marginalization of the ethical it implied.

According to Murdoch’s analysis, the motivation for separating facts and value is to keep value “pure and untainted, not derived from or mixed with empirical facts” [13] (p. 25). We assume that facts are objective and values subjective and, since it is the former kind of conclusions we want, we need to abstract facts from any subjective value elements. The segregation may have been well intentioned, Murdoch writes, but it ignores “an obvious and important aspect of human existence, the way in which almost all our concepts and activities involve evaluation” [13] (pp. 25–26). The very concept “fact” is set up by human agents and is accordingly not a neutral concept. This means that the supposed neutrality that the division between facts and values is supposed to uphold is fictitious. Values pervade and colour what we take to be the reality of our world, Murdoch writes [13] (p. 26). In other words, *everything* has a moral character (“‘But are you saying that every single second has a moral tag?’ Yes, roughly” [13] (p. 495), and—as has since been recognized—surveying facts will always involve some kind of moral discrimination. I will fill out this sketch and its implications below, but for now, the take-away lesson is that facts and values cannot be divided into two separate domains, where facts are unaffected by values. This makes it difficult to see what an interaction between them should look like. This, however, is precisely the problem Molewijk et al. set out to solve.

## 6. What Is Being Integrated in Integrated Empirical Ethics?

As already mentioned, empirical bioethicists recognize that we have to pay attention to the ways in which values are negotiated and contingent on particular circumstances that are shaped by material and social structures. Molewijk et al. do too: their framework takes as a guiding principle that there is no fundamental distinction between fact and value. They defend what they term an *integrated empirical ethics*, which aims to “bridge the normative–empirical chasm and rejects the very idea of distinguishing between purely normative and purely empirical claims” [5] (p. 58). This is described as studies where “ethicists and descriptive scientists cooperate together, integrating moral theory and empirical data in order to reach a normative conclusion with respect to a specific social practice” [5] (p. 57). But even though their approach has many merits, I want to argue that they do not successfully leave behind the metaphysical outlook Murdoch criticized. This is due to two reasons. First because of how the relationship between fact and value is conceived and second because of the kind of justification the integrated normative output is thought to give.

To examine the first claim, we need to ask what it means to say that there is no fundamental distinction between fact and value. It does not mean, as Molewijk et al. note, that these terms are synonymous; it merely means that they are “mutually constitutive”. They rightfully acknowledge that “[t]here is no Archimedean point of view (i.e., one truth) from which all scientists can acknowledge the world as it is” [5] (p. 58), but they maintain that the main question remains to figure out what the interaction between the normative and the empirical consists in and how it can be carried out. For example, they discuss two possible methodological candidates—pragmatic hermeneutics and reflective equilibrium—which both aim at guiding the empirical bioethicist researcher in finding the right balance between empirical input and normative conclusions. They also comment on when this interaction takes place; while this will ultimately depend on the specific goals of the project, an interaction will always happen during the project planning and data collection and during the final phase of drawing normative conclusions [5] (p. 63).

The very question of how an interaction between the empirical and the normative can be achieved reveals just how difficult it is to move beyond concepts and worldviews that have such as strong hold of us as the fact/value distinction does. To say of two domains that they are mutually constitutive is precisely to acknowledge that there are two domains, and to ask how and when an interaction happens is to assume that we can use our theoretical reasoning to draw the borders between them.

When we consider how researchers deal with these distinctions in practice, we can actually find some reasons why they may not be particularly apt, something which is revealed in Molewijk et al.’s article. Even though many articles that are written in empirical bioethics start by acknowledging the problems associated with the fact/value distinction, in practice, a study of 33 empirical research projects found that less than half contained information on their relationship [5] (p. 63). Those that do, furthermore, are unclear about the specific features of that relationship and researchers can rarely say whether something is purely empirical or purely normative. In addition, Molewijk et al. remark, with some surprise, that most researchers endorse the assumption that a moral statement cannot be deduced from merely empirical statements, while only a few of them actually report to have experienced problems with the is–ought distinction.

I want to suggest that this, too, is telling for the kind of grip our concepts can have on us. If few researchers can clearly distinguish what is empirical and what is normative or specify the features of that relationship, this is perhaps not because we have not yet developed the appropriate methods for doing so, but because the frameworks ask the wrong question. Furthermore, if few researchers writing within empirical bioethics do not experience the is–ought distinction as challenging in practice, this is perhaps because explaining what happens in that research process as a kind of illegitimate leap from a description to a normative statement does not accurately describe what takes place. In the following, I will argue that Murdoch and Diamond can offer us a better explanation.

## 7. Murdoch’s More Radical Challenge

Above, we saw that Murdoch argues against the fact/value distinction in that it cannot be maintained because values are, in fact, pervasive. One way this claim can be interpreted is that there can be no fundamental distinction between facts and values. However, Murdoch’s point is rather that there is an important distinction between them, just not the one that is commonly assumed. I turn to Diamond’s analysis of Murdoch to explain this point.

Diamond suggests that we should understand this claim by considering what it means for something to be an object for cognitive activity. A field such as history can be an object for cognitive activity in the sense that it consists of statements that are explained, justified and backed up with plausible evidence according to a set of standards that are specific to this discipline [12] (p. 106). But while history and other humanistic or scientific disciplines can have such standards, the same does not hold for values. The difference between facts and values, Diamond elaborates, is that history or botany, for example, are not ubiquitously present; it is something we can enter in and out of engagement with, investigate and analyse. However, values, in contrast, are ubiquitous, and hence we cannot engage with them in the same way or aim to put them under scrutiny through similar standards [12] (p. 107). Saying that values are ubiquitous is precisely an act of distinguishing value from whatever can be a subject matter for consciousness.

The pervasiveness of values means that ethics cannot be a branch among other philosophical branches. Murdoch warns us against carving out ethics as a separate field in this manner and argues that the boundaries we draw for ethics as a field of study usually reflect a moral outlook [14] (p. 53). Diamond brings this out in the following quote: “If value is said to be ubiquitous, this is in fact tied to the way in which our experience of the world can bear morally on any situation, can shape our vision of what a situation is” [12] (p. 108). This understanding has implications for how we understand moral thinking more generally.

The concept “vision” plays a major role in Murdoch’s thought. Up against the idea of morality as a choice we make based on the facts we have established, Murdoch defends the idea of morality as a question of vision, of seeing things in a certain way. Philosophers, Murdoch writes, seek universal rules, but when we leave this ambition behind, “we come into the cloudy and shifting domain of the concepts which men live by” [15] (pp. 74–75). Murdoch’s criticism of the logical positivist way of thinking was that it assumes a world that is given prior to moral thought and life [14] (p. 56). This is a misleading picture for many reasons, among others because it leads to an oversimplified conception of why people think differently, Diamond writes. There is no one way to understand the relationship between our concepts and reality and, furthermore, the idea of concepts as being cloudy and shifting implies that these ideas are prone to change. Murdoch encourages us to “resist the temptation to unify the picture by trying to establish, guided by our own conception of the ethical in general, what these concepts *must be*” [15] (p. 75). Instead, Diamond notes, Murdoch’s idea is that it is through elaborating and applying moral concepts that we come to understand “what the world is, what life is” [14] (p. 57). In the following, I will argue that much of the research that takes place under the umbrella of empirical bioethics can be understood as a congenial way of doing so. This will require that we return to the question of what it may mean to learn from experience and how that question has been dealt with philosophically.

## 8. Taking Empiricism Back from the Empiricists

In an unpublished paper titled: “Murdoch Off the Map, or, Taking Empiricism Back from the Empiricists”, Diamond defends what she calls a *reflective empiricism*, which she describes as an “empiricism free of the demand that to take the empirical seriously in philosophy, one must engage with the empirical sciences” and as “an empiricism of reflection on human experience” [16].

Diamond’s statement that she wants to defend an empiricism free of the demands that philosophy should engage with the empirical sciences may, at a first glance, make her an odd candidate to turn to for a defence of the relevance of empirical research for bioethics. However, her aim is not to argue that it is irrelevant for philosophers to engage with empirical sciences or that doing so would be detrimental to philosophy. The target of Diamond’s criticism has rather to do with a conception of the work this research is supposed to do in philosophy, the assumed standard it has to pass, and what she takes to be a narrow-minded understanding of which methods are available for philosophers. Making this point requires clarifying what we mean by both the terms “experience” and “empiricism”.

There is an obvious sense in which experience is the basis of all philosophy, and there is an obvious sense in which philosophers bring experience into their philosophizing. However, there are questions about *how* experience should be brought into philosophy and there are questions about *how large* the experiential basis needs to be. One of the claims that seems to underlie the social science critique, or at least how bioethicists have understood and taken up this critique, seems to have to do with insufficient experience basis: philosophers who engage in abstract normative analysis are accused of having a basis that is too thin. This is why we should talk to stakeholders, patients or laypeople about the issues we want to address, so that we can have a broader and better basis for making claims.

To take an example of a way of bringing experience into philosophy that Diamond is critical of, we can look to experimental philosophy. Experimental philosophy, too, has called for a philosophical turn to the empirical; they criticize armchair philosophers who merely rely on their so-called ‘intuitions’ when they want to make claims about the world and argue that we should proceed scientifically to back up our claims. Hence, they use empirical data about ordinary people’s intuitions, often collected through surveys, to inform research on philosophical questions [17]. This kind of quantitative opinion polling, however, is not what the empirical bioethicist is after. Empirical bioethics is concerned with understanding the contexts in which bioethical issues arise and are fought out and with having this understanding inform philosophical analysis. This is a different aim, and one that goes a long way in being sensitive to the Murdoch/Diamond line developed here. However, both enterprises rely on the kind of empiricism Diamond wants to challenge as a model for philosophy. They seem to assume that the philosopher has the choice either between relying on so-called ‘intuitions’ or engaging with the sciences and adopting a scientific method, and both suggest that it is the latter that will grant answers. “Philosophical thought”, Diamond writes, “that draws on human experience but not in a scientific or protoscientific way seems not to have a place in this conception of the range of philosophical approaches.” [16]. What Diamond wants to discuss is what this way of engaging with experience means and how it could be seen as a possible method in philosophy. Reflective empiricism, for Diamond, designates this humanistic method of thinking about experience.

In typical Diamond fashion, a more precise definition is not offered, and it can be difficult to understand what this method consists of—few clues are given as to how it could be applied in practice. In elaborating, however, she uses the phrase “things that are good to think with” [16], which I find to be a useful term for understanding Diamond’s broader conception of empiricism. She intends this term to capture a broad category of notions, concepts, stories and ideas that guide how we approach and understand the world. Part of the humanistic enterprise, in her view, is to revise the things that we use to think with. Diamond quotes a description Geoff Dyer has offered for this kind of learning, where he talks about “that ongoing thing of the way we learn stuff through literature, the humanities”. In this sense, “[p]hilosophy is not more “empirical” in engaging with experimental studies […] than in engaging with Shakespeare” [16].

Dyer has fiction in mind, and Diamond’s examples also tend to be literary. Works of fiction appeal to the imaginative faculties and function as a site where our understanding of moral issues can be challenged and reimagined. One example she gives is Bernard Williams on how we can learn from the ancient Greeks. When Williams discusses why we learn and think about the Greeks, “he notes that they don’t really tell us about themselves, they tell us about *us*” [16]. The Greeks’ understanding of human agency, responsibility, regret and necessity can illuminate our own understanding and perhaps show us what is missing in it. In other words, these stories can be good to think about such questions *with*. They can help us to examine and revise the concepts and narratives with which we approach different situations.

The transformative forces of engaging with narratives are not limited to fiction, however. What I want to suggest is that the kind of qualitative research that is being conducted in empirical bioethics can serve the same or similar functions. This is an argument Hämäläinen makes in her article “Wittgenstein, Ethics and Fieldwork in Philosophy”. So far, this article has been concerned with philosophers’ turn to the empirical. However, in many philosophical subfields, applied or empirical ethics is regarded with some suspicion. As Hämäläinen comments: “there is a readiness to suspect anyone who does so [engage with the empirical] has misunderstood what philosophy is properly about” [18] (p. 38), implying that real philosophy happens in the armchair. But Hämäläinen points out that in disregarding the empirical, we might actually miss out on very useful resources for our thinking about philosophical questions:

When people bring up empirical objections in philosophical conversations they are not necessarily enthralled by some wrongheaded conception of the all-saving power of the empirical, nor making a category mistake. They might actually be doing something that is quite congenial to Wittgenstein’s conception of philosophy as a struggle to understand our own form of life and the concepts we live by [18] (p. 39).

Fieldwork, as Hämäläinen suggests, aims to uncover the reasoning and values that are inscribed in the way we use language. It can function as a site for exploring moral choices and concepts and contribute to opening up our forms of life to investigation. In the next section, I will discuss what I take to be an exemplary case of this, namely Jonathan Ives’ article “Encounters with experience”.

## 9. Example II: Ives’ Encounters with Experience

Ives’ article on bioethics as encounters with experience is an exemplary account of the kind of attention to experience Murdoch and Diamond encourage. “Encounters with experience”, Ives writes, sum up what empirical bioethics is all about. He elaborates: “It is about philosophers ‘getting their hands dirty’, getting out of their Platonic ivory towers and acknowledging that ethics is about people, not just good arguments” [19] (pp. 2–3).

To illustrate what he means by this, he uses his own experience from his doctoral research, which explored the moral significance of father/child relationships. At the outset of his research, Ives explains, his supervisor joked that the best way to examine this issue would probably have been to have a child with his wife, have an affair and become pregnant with a lover and refuse to pay child support and then move in with another woman and raise her children by another man as his own, providing him with first-hand experience of such relationships [19] (p. 3). However, an essential point in empirical bioethics is that we can draw on resources outside of our own experience to inform our moral thinking. In Ives’ case, this was done through a series of focus groups. In an illuminating passage, he details what he learned from this experience:

It has enabled me to examine, in depth, the reasons men have for feeling the way they do, and the values they hold that inform their views. It has sensitised me to the realities of fathering, brought me into contact with a range of perspectives and emotions that I would otherwise not have encountered, and it has enabled me to begin to produce a philosophical framework that is sensitive to those realities [19] (p. 3).

The phrase “bringing into contact with a range of perspectives and emotions that I would otherwise not have encountered” is important here. It says a lot about the value of being empirically informed generally, but it also says something more profound about how having such conversations and being exposed to new narratives can change our perspectives. Ives writes about how, initially, he had the conviction that fatherhood had nothing to do with biology and should be conceived of only in social terms. As he began to conduct the focus groups and started to have conversations with men and with fathers, however, he “began to realise that it could not be so simple” [19] (p. 3). Instead of understanding fatherhood as a social construct, he started to think that there are in fact two understandings of fatherhood, one genetic and one social. He was further able to gain an understanding of the significance of these relationships and found that while social fatherhood is valued more than genetic fatherhood, genetic fatherhood is often thought of as facilitating a better social relationship.

The encounters that Ives had made him sensitive to this distinction and its significance, and he writes that he would not have come to think of it this way without them [19] (p. 3). Ives, then, has a very clear sense of the kind of value empirical research can have for philosophical endeavours. What he gives is actually a very good description of what happens when we come to see something in a new light, as Murdoch describes it, and improve our concepts to make them more responsive to experience. For Murdoch, it is when we explore, elaborate and apply moral concepts that we come to understand the world [14] (p. 57). The concepts themselves are essential in giving the character of the world—they are deep moral configurations of the world [14] (p. 58). In his own attempt to situate the field, Ives describes it as neither being descriptive ethics nor strictly normative ethics, but as “chimerical hybrid, combining elements of both—a mythical ‘research monster’ with the head of a philosopher, the tail of a social scientist, and a body of exchangeable composition that is most likely determined by its native soil” [19] (p. 3). But we need not think of it as a research monster at all. On the contrary, what Ives is describing is something as mundane as drawing on resources outside of his own experience to help him think better about a specific topic. When Ives comes to see the concept of fatherhood differently, it is because he has become attentive to the realities of what being a father can be like.

## 10. Salmon Revisited

While empirical projects such as the one conducted by Ives are gaining ground, particularly within medical ethics, there have so far been few attempts to use such methods in discussions on animal ethics [20] (p. 167), [21] (p. 854). While we can ask human patients about their preferences and values, these are obviously meaningless question to ask animals. However, as Persson and Shaw convincingly argued, the arguments for using empirical research/data in medical ethics apply to animal ethics too [21] (p. 856). As they remark, methods from the social sciences can be used to find out whether a norm has been successful: “It leads away from questions like ‘can they suffer’ or ‘does it matter whether their suffering can be compared to ours?’ and towards questions like ‘is it morally acceptable what we do in line with our norms/laws to those creatures which certainly do suffer’” [21] (p. 858). The former questions are what traditional approaches to animal ethics, such as the ones defended by Peter Singer and Tom Regan, are concerned with. Singer argues that the relevant question to ask in animal ethics is whether or not an animal experiences harm. He defends equal consideration of interests, that is that equal interests must be given equal moral weight, and argues that we thus must assume species impartiality [22]. Regan, for his part, takes as his central concept “rights” and argues that non-human animals have moral rights because they are “subjects-of-a-life” [23]. However, in the case of using genetic engineering on animals, neither of these concepts seem able to capture what is at stake, since concerns move beyond welfare and health, as de Vries argued [20] (p. 5). A few examples he gives that are confirmed in my own material are that genetically engineering animals, especially in the case of transgenesis, is unnatural and therefore morally wrong; that it amounts to playing God and is an expression of human hybris; and that using genetic engineering further instrumentalises the lives of animals [20] (pp. 5–6). Concepts of suffering and equal considerations of interests do not seem suited to address these concerns [20] (p. 6). In the case of salmon, genome editing can be used to significantly decrease their suffering, for instance by making them resistant against salmon lice and diseases, which are a major welfare problem. However, we seem to sidestep or miss part of the picture of what is at stake when we evaluate this potential use of the technology if we reduce the question to suffering and pain. De Vries makes another salient point: Singer’s theory states that we ought to consider the interests of animals, but what if these interests are altered using gene editing technology? [20] (p. 6). Singer seems to be unable to say something about this. In a similar manner, Regan’s idea of animals as subjects of a life gives us no clue as to how to address genome editing of embryos, as they are not such subjects yet [20] (p. 6).

Diamond argues against the notions of capacities and rights in animal ethics and points out that we can never decide how to act towards an animal based on what kind of characteristics they have. Rather, animals are our ‘fellow creatures’ [24,25]. She writes that this is an idea of “being in a certain boat, as it where, of whom it makes sense to say, among other things, that it goes into Time’s enormous Nought, and which may be sought as company” [24] (p. 474). Importantly, this idea depends upon “a conception of ‘human life’” [24] (p. 474). For Diamond, concepts such as “human” or “animal” are not biological concepts, they are conceptual configurations that enable us to make sense of the world. This is “the cloudy and shifting domain” Murdoch writes about that was referred to above. For instance, a laboratory mouse, a pet mouse and a wild mouse have more or less the same capacities and characteristics, but they each give rise to a different set of norms and considerations of what kind of treatment is considered acceptable. Which values are at play in a given situation are constituted through the forms of life we participate in. Diamond argues that our idea of animals is not given to us independently of our ways of thinking about and responding to them [24] (p. 476). This opens up a space for exploring practices, values and traditions.

Just like Ives was jokingly encouraged to enter into various father relationships in order to get a better understanding of fatherhood, one way for me to gain insight into salmon practices would have been to get a job on a salmon pen or follow hobby fishers of wild salmon into the rivers, for example. I have fished (ineptly) in the past, but that about covers the extent of my experience and knowledge about fish. Fish are not animals that I have expressed any particular concern for and it was chance rather than interest that had me end up on a salmon project. My perception of salmon has, however, gradually changed during the course of interviewing and learning more about them. Initially, when my supervisor would describe salmon as ‘iconic’, I thought it exaggerated and odd, but as similar concepts, such as ‘beautiful’ and ‘amazing’ have been invoked to describe salmon (in particular, the wild variant) and as I have come to understand the significance of salmon in Norwegian and Samí culture and the respect and love for them that many who deal with them have, this has altered how I think about them. In this paper, I cannot discuss the data and their implication for the question on genome editing, though I plan on doing so elsewhere. One thing that gradually became clear over the course of the interviews was how different and occasionally conflicting notions of responsibility come into play with regard to wild and farmed salmon. If we take, for example, the paradigmatic case of sterile salmon, the motivating force behind introducing it into production is to preserve the wild salmon. However, the main concerns about genetically engineering salmon so that it becomes sterile also have to do with preservation: there are concerns about whether being sterile could affect the farmed salmon negatively in some way and there are concerns about other kinds of negative impact an escaped sterile salmon could have on ecosystems. Finally, there are concerns about how it affects us as moral agents to adapt animals to increasing extents to make them fit production purposes and where we draw the border for what kind of interventions we make. In addition to good arguments and reasons for why we should do or avoid doing something, such descriptions have served a very particular purpose in that they have made me view fish differently and understand what kinds of attitudes and values come into play when dealing with them.

## 11. Conclusions: Empirical Ethics against the Tornado Model

This article attempted to defend reflective empiricism as an approach to empirical ethics that does not fall prey to the integration problem and argue for the relevance of this approach in animal ethics. More needs to be said about how this can be applied in practice and particularly about the relationship between this approach and social science methodology and what is required of philosophers who do this kind of work. That, however, is beyond the scope of this article.

In conclusion, I want to return to the question of the relationship between knowledge and experience. When Diamond challenges the standard conception of empiricism, what she challenges is not an engagement with the empirical as such, but rather what she takes to be a narrow standard of what it can mean to use experience. Elsewhere, when Diamond writes about this standard, she refers to it as ‘the tornado model’ [26]. If we want to contribute to scientific knowledge about tornados, instruments seem like a more reliable source of information than people: we want to understand how tornados work and not necessarily what it is like to be in a tornado. People will simply lack the accuracy for knowledge production in this case and the relevant experience is experience with reading the instruments that produce knowledge. What Diamond argues is that this model of knowledge has an impact on other areas of knowledge production, downplaying the relevance of other kinds of experience, favouring emotional distance between the investigator and the investigated [26] (p. 1007). There are, however, a variety of areas in which we might miss out on crucial knowledge if we let the tornado model dominate, and Diamond turns to animals as an example. Often, Diamond writes, people’s experiences with animals are dismissed in scientific discussions about them on the account that they are mere expressions of emotional states. When we want to say something about how animals ought to be treated, we are not interested in such sentimental responses; rather, what we are looking for are some objective criteria about moral status expressed in an as neutral language as possible. But in this case, Diamond argues, the tornado model “fails to allow for an understanding of what animals can do that grows out of working actively with the animals in a way inspired by love and respect and even a certain humility.” [26] (p. 1008). To this, I would add that it also fails to allow for an understanding of what we can do to animals. Using qualitative research methods to attend to such experiences can shed light on our moral relationships with animals, which has normative implications for how we ought to treat them.

## Data Availability

Not applicable.

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
