# Peer review of "Reflective Empiricism and Empirical Animal Ethics"

_animals, 2022, doi:10.3390/ani12162143_

Round 1

Reviewer 1 Report

This is a refreshingly well written, engaging paper with interesting examples (the supervisor's comments to the fatherhood researcher - wow!) and integration of work with salmon. I tend to do more applied research and this was quite theoretical, but I think there is enough interest in empirical ethics to warrant this discussion. Usually my review reports are pretty constructively critical but I cannot really fault this - well done to the author.

My suggestions are extremely minor and I don't need to review again:

Line 77 - study instead of studied; cited instead of sited

Line 565-569 - I would write "farmed salmon" rather than "the farmed salmon", and "wild salmon" rather than "the wild salmon"

Line 569 - becomes

Line 581-583 - what more needs to be said and when? Is there another way of phrasing this ie does it need to be further explored?

599-606 - reading this passage made me think of the book "Learning Animals" by Nadine Dolby, who argues that veterinary education strongly socialises/indoctrinates veterinary students  away from thinking about and talking about love for animals. It might be worth a read if you have the time.

Author Response

Thank you for taking the time to read and provide feedback on my article, and thank you for the very encouraging words. I have implemented the suggested language edits. Unfortunately, time did not permit me to read Dolby, but I've made a note for the future! Thanks again for your time and extremely kind feedback. 

Reviewer 2 Report

I believe  as an animal rights philosopher and an individual who supports universal ethics in terms of human and animal rights I have a bias against the paper as it tends to support a sort of cultural relativism.  I am not an armchair philosopher as I am heavily involved with multiple aspects associated with non-human animals. Therefore, my review should perhaps not be considered. The paper is well written but I disagree with the underlying assumptions.

Author Response

Thank you for taking the time to read my article. My understanding is that you are abstaining from giving feedback since you feel biased against it, so there are no revisions to consider here. I would like to note that I do not assume cultural relativism, and this is also not Cora Diamond's position. 

Reviewer 3 Report

This is a really interesting and well written paper which makes an important contribution to scholarship in empirical bioethics by injecting new ideas and conceptual resources (Murdoch and Diamond) into the discussion.

A couple of small suggestions – footnote 1 gives a number of instances of ‘turns’ in philosophy. Given the paper touches on animal philosophy, consider referencing the political turn in animal ethics.

Readers may not be familiar with the Samí; perhaps a footnote explaining they are a people indigenous to Scandinavia.

A more substantial point; there are a number of interesting threads running through the paper e.g. about empirical bioethics, the philosophies of Murdoch and Diamond and genetic modification of salmon, but the latter is not developed in sufficient depth. Given the paper has been submitted to the journal ‘Animals’ this needs to be remedied. The author might consider deleting some/all of the section on Ives and fatherhood in order to devote greater attention to the salmon example.

There are a number of issues with written expression that need to be addressed, some are given below.

‘Progress’ rather than ‘proceed’ in the sentence on pages 30-31.

No need to list all the authors of the paper “Empirical data and moral theory: A plea for integrated empirical ethics” on page 74.

In the sentence on page 78 ‘I argue that although the authors acknowledge the interdependence between facts and values’, it should be ‘of’ rather than ‘between’.

Delete the comma after ‘diseases’ in ‘A few such examples include salmon lice and other diseases, severely impacting the life quality of the farmed salmon, and the fact that they escape from the pens in large numbers, breeding with the wild populations.’

In the sentence of page 153, consider deleting ‘we should’ in ‘A reasonable question that might be raised is why we should bother to talk to these groups at all’.

Page 177 “Both undertakings are worth commenting,’ requires ‘on’ before the comma.

Line 215 rather than ‘no longer has the standing it used to,’ ‘no longer has standing it once did’.

Line 224 ‘was in the air’ is too colloquial.

Page 245 ‘I will fill out this sketch of this view and its implications below’ remove 1 of the two instances of ‘this’ in the sentence.

Page 247 in ‘unaffected from values’ insert ‘by’ rather than ‘from’.

Page 263-264, avoid ending the sentence with a preposition ‘normative output is thought to consist in.’

Page 265, ‘To examine the first claim, we need to ask first what it means to say that there is no’ avoid repeating ‘first’.

Delete ‘they note’ page 290.

Page 306 ‘Above, we saw that Murdoch argues against the fact/value distinction that it cannot be upheld because values are in fact pervasive’, perhaps ‘maintained’ or similar rather than ‘upheld’.

Page 307 ‘One way this claim can be interpreted is that there can therefore be no fundamental distinction between facts and values’, reword to avoid ‘there’ and ‘therefore’ in close proximity.

Page 321 should be ‘them in the same way’, i.e. currently missing ‘in’.

Page 377 ‘To take an example of a way of bringing experience into philosophy that Diamond is critical towards, we can look to experimental philosophy’ should be ‘critical of’.

Page 511 rather than ‘Regan, on his end’, ‘for his part’?

Author Response

Thank you for taking the time to read and provide feedback to my article! 

I have implemented the grammatical changes which were suggested, thank you for this. I have also included a sentence about the Samí and a reference to the political turn in animal ethics.

I understand your point about it being desirable to develop the final section on salmon further and remove the example from Ives. However, I have not done so in this resubmitted manuscript for four reasons. First, one of the other reviewers highlighted the Ives part as a positive contribution. Second, I think it is important to keep the section on Ives' research on fatherhood to show how the method for which I am suggesting a theoretical framework is actually exemplified in the existing literature on empirical bioethics. Third, I think using an already existing, well-known example is a point in itself and an efficient way to show how there are many good practices out in the field which need not give rise to the meta-philosophical problems it is often preoccupied with, and it is also an efficient way of entering into dialogue with that literature. Fourth, the text is already long, and developing the salmon part further would make it even longer. I agree that there is more to do on that account, but I hope to continue these reflections in a follow-up paper.

Thanks again for your comments and for your encouraging feedback!